# Blood Vessel Imaging at Pre-Larval Stages of Zebrafish Embryonic Development

**DOI:** 10.3390/diagnostics10110886

**Published:** 2020-10-30

**Authors:** Alexander S. Machikhin, Mikhail V. Volkov, Alexander B. Burlakov, Demid D. Khokhlov, Andrey V. Potemkin

**Affiliations:** 1Laboratory of Acousto-optical Spectroscopy, Scientific and Technological Center of Unique Instrumentation, Russian Academy of Sciences, 117342 Moscow, Russia; machikhin@ntcup.ru; 2Department of Applied Optics, University ITMO, 190000 Saint Petersburg, Russia; ph-m.volkov@yandex.ru (M.V.V.); appanpotemkin@gmail.com (A.V.P.); 3Department of Ichthyology, Faculty of Biology, Lomonosov Moscow State University, 119991 Moscow, Russia; alexander_burl08@rambler.ru

**Keywords:** zebrafish, embryonic development, cardiovascular system, in vivo imaging, optical mapping, non-invasive measurements

## Abstract

The zebrafish (*Danio rerio*) is an increasingly popular animal model biological system. In cardiovascular research, it has been used to model specific cardiac phenomena as well as to identify novel therapies for human cardiovascular disease. While the zebrafish cardiovascular system functioning is well examined at larval stages, the mechanisms by which vessel activity is initiated remain a subject of intense investigation. In this research, we report on an in vivo stain-free blood vessel imaging technique at pre-larval stages of zebrafish embryonic development. We have developed the algorithm for the enhancement, alignment and spatiotemporal analysis of bright-field microscopy images of zebrafish embryos. It enables the detection, mapping and quantitative characterization of cardiac activity across the whole specimen. To validate the proposed approach, we have analyzed multiple data cubes, calculated vessel images and evaluated blood flow velocity and heart rate dynamics in the absence of any anesthesia. This non-invasive technique may shed light on the mechanism of vessel activity initiation and stabilization as well as the cardiovascular system’s susceptibility to environmental stressors at early developmental stages.

## 1. Introduction

The zebrafish has emerged to become a common and useful vertebrate animal model for cardiovascular research in recent years [1,2,3,4]. It is a powerful genetic system to study cardiac function due to several advantages. The ability to implement the main modern genome editing tools and techniques, including engineered nucleases (meganucleases, zinc-finger nucleases, transcription activator-like effector nucleases) and CRISPR gene editing, have been demonstrated for *Danio rerio* [5,6,7,8]. Various zebrafish transgenic lines can be used to enhance the visualization contrast of different transformation processes [9,10,11]. The use of the zebrafish as a model organism is of particular interest for the analysis of innate immunity [12] and for the study of viral [13,14,15], bacterial [16,17] and fungal infections [18,19].

Small size, ease of maintenance, short reproduction cycle with multiple (up to 2000) embryos and short period (3 days) of embryonic development make *Danio rerio* a convenient object for continuous observation and experimental research of the mechanisms by which genetic and disease-related modifications are being passed to the offspring.

Cardiovascular system functional characteristics of zebrafish are well studied in various conditions and in the whole life cycle. Their accurate analysis and patterning may indicate developmental disorders and presence of pathology at the very early stages. Detailed studies of cardiovascular system function are of great value for understanding cardiac failures and identifying novel therapies for human cardiovascular disease [20,21,22].

Zebrafish embryos develop in the external environment and are relatively small-sized and transparent in the optical wavelength range. These features allow one to easily visualize the circulatory system structure [23,24], observe heart and blood vessel formation and development, and simulate cardiovascular diseases [25,26,27,28]. While the zebrafish cardiovascular system functioning is well examined at larval stages [24,29,30,31], the mechanisms by which the vessel activity is initiated under natural conditions remain a subject of intense investigation. At the pharyngula stage (24–40 hpf) that precedes hatching [32], invasive procedures (excising, fixation, sectioning and staining) are not effective for real-time monitoring due to the fast-changing and vulnerable state of the emerging cardiovascular system.

Thus, biomedical optical imaging techniques are remaining promising in the field of in vivo zebrafish studies [2,33]. Though heart function analysis from bright field time-lapse image sequences [31,33,34,35] as well as quantitative research of zebrafish heartbeat initiation and stabilization [36] have been reported, blood vessel imaging during early embryonic development has not been carried out. Due to constant embryo motion within the shell, conventional processing algorithms based on image subtraction are inefficient and require precise initial image stabilization. In this research, we show that a combination of time-lapse bright-field microscopy and an advanced image processing technique represents an effective approach to in vivo stain-free cardiac activity mapping across the whole specimen and quantitative analysis of cardiovascular performance even at pre-larval developmental stages in the absence of any anesthesia.

## 2. Materials and Methods

### 2.1. Experimental Animals

Zebrafish embryos were from an existing stock at the Biological Faculty of Lomonosov Moscow State University. Before embryo collection, single species groups of males and females were kept by local aquarists in isolated 10 L glass aquaria at a temperature of 26 °C with the aquaria illumination turned on for 12 h daily. The selected groups were fed three times a day ad libitum on flake and *Artemia*. For embryo collection, males and females were placed together in a breeding tank and the embryos were collected immediately post-fertilization. A mixture of individuals from several different breeding groups was used during the experiment to maximize genetic variation among embryos. The two-cell development stage was used as initial time point during observation because the exact time of fertilization was difficult to determine. The development, survival rate and morphology of the collected embryos was not affected in comparison to the control group not subjected to the study.

### 2.2. Experimental Setup

The conventional off-the-shelf trinocular transmitted light bright-field microscope is the basis of the experimental setup (Figure 1). A Koehler system was implemented to achieve uniform illumination of the object. Images were formed by an optical system assembled of a flat field corrected apochromatic objective (10× NA 0.25) and a standard tube lens. To acquire the digital images, the complementary metal-oxide-semiconductor active-pixel image sensor (IDS uEye UI-3060CP-M-GL Rev.2, 1/1.2”, 1936 × 1216 pixels) was installed onto the camera tube. The image sensor was connected to the PC for raw data acquisition and storage. The image processing and data analysis pipeline is described in Section 2.3.

### 2.3. Experimental Protocol

The single experimental dataset contained 3000 12-bit grayscale digital images with 1200 × 1200 pixel resolution obtained during a 60-s time series (50 fps). For preliminary geometrical calibration of the imaging setup, the images of the test chart were captured before the experimental dataset acquisition. The geometrical calibration procedure is an essential step for the magnification determining and estimation of the real image distortions. During image acquisition, the inspected embryo may move and rotate within the shell, and illumination conditions and background characteristics may also vary. Data analysis procedures require pre-processing of raw images to crop the region of interest, compensate for movement and eliminate the illumination non-uniformity (Figure 2). After these procedures, we obtained the well-matched and intensity-corrected spatiotemporal data cube *I*(*x*,*y*,*t*) including time-domain dependences *I*(*t*) for each image pixel with spatial coordinates *x*,*y*.

Figure 3 illustrates the main image pre-processing stages shown in pink color in Figure 2. First, the microscopic images of 26 hpf unhatched zebrafish embryo in a lateral position were to be cropped down to the embryo dimensions (Figure 3a). Second, to compensate for illumination non-uniformity and align the average intensity of all images in a series, we subtracted a smoothed image and constant term 127 (Figure 3b). Third, to ensure pixel-to-pixel matching of all images, we calculated local motion vectors and matched the images with respect to their directions and lengths. Figure 3c shows a locally matched image with crosses indicating the positions of each grid knot in the first (black) and the last (white) image of the sequence. Shape and intensity of the temporal signal *I*(*t*) in each pixel of the obtained data cube *I*(*x*,*y*,*t*) characterizes morphogenetic events occurring during embryonic development. The blood flow changes may be detected by the periodic variations of the transmitted light intensity since blood cells absorb light more strongly than the surrounding tissues [37]. Plotting the spectrum of the temporal signal using Fourier transform allowed detection of the dominant frequencies, which corresponded to the cardiovascular activity. After obtaining the intensity deviation values in each image pixel, we could subtract the blood-free background and consider only the pixels with blood flow related to significant intensity oscillations. Thus, we obtained a series of well-matched blood flow images ready for cardiac activity analysis (Figure 3d). The blood circulation was clearly visible throughout the full cardiac cycle (Appendix A).

## 3. Results

From the series of blood flow images (Figure 4a), we could calculate the intensity of blood volume changes as the ratio of high- and low-frequency spectral components in the Fourier spectrum. Thus, we obtained the vessel image. It vividly depicts the tissues associated with cardiac activity, i.e., the spatial structure of the cardiovascular system existing at this developmental stage (Figure 4b). The heart and the system of vessels that carry blood throughout the embryo’s body may be clearly identified on the blood-free background [32,38,39]. The calculated vessel image in Figure 4b demonstrates the efficiency of the proposed processing algorithm and shows the main elements of the existing cardiovascular system. Vessels are named according to commonly used classification [40]: ACeV—anterior cerebral vein; BA—basilar artery; CA—caudal artery; CaDI—caudal division of the internal carotid artery; CCV—common cardinal vein; CV—caudal vein; DA—dorsal aorta; MsA—mesencephalic artery; MsV—mesencephalic vein; PCV—posterior cardinal vein; PHBC—primordial hindbrain channel; PHS—primary head sinus; PICA—primitive internal carotid artery; PMBC—primordial midbrain channel; PMsA—primitive mesencephalic artery; PPrA—primitive prosencephalic artery.

Besides the two-dimensional mapping of cardiac activity, the obtained spatiotemporal data cube *I*(*x,y,t*) allowed calculation of the heartbeat rate and blood flow velocity—quantitative parameters characterizing cardiovascular functioning. The heart location (Figure 4c) could be detected automatically as the image pixel group demonstrating the most intensive and constant oscillations. Heart area detection in the blood flow images allowed accurate cardiac beat detection and heart rate monitoring. The obtained temporal signals in the heart area had the correct shape and spectrum [30,34]. In the experiment shown in Figure 4, the measurement result for the heart rhythm was 69 bpm (Figure 4h), which is in good agreement with the values obtained in other studies for this stage of zebrafish development [36,41].

Pulsatile flow of blood cells could be easily detected in zebrafish embryo blood vessels from the beginning of circulation at 24 hpf. To demonstrate blood flow velocity measurement, we selected one of the vessel areas (Figure 4d) and carried out its morphological analysis in order to detect the offset vector field (Figure 4e) and central line of the vessel (Figure 4f) and to calculate normal direction at each point. Implementation of this procedure allowed the alignment of the pixels corresponding to the blood vessel into a straight line [42] to calculate the blood flow velocity. Since the blood vessel pixels were aligned, the relative shift between consequent images could be estimated. Then the blood flow velocity could be determined as the ratio of this shift to the time interval between the moments of image acquisition. The obtained blood flow velocity signal was compared with the temporal signal in the heart area to validate the proposed velocity measurement technique. In the diagram in Figure 4g, the curves corresponding to these signals are superimposed so that their mean values are equal. Heart area temporal signal and blood flow velocity signal had the same period of 0.86 s and a constant relative temporal shift of 0.35 s. Average blood flow velocity was 389 μm/s, which is close to the value measured using quantitative fluorescent imaging at the larval stage (Figure 4i) [24].

We implemented the described image processing algorithms in C++ using parallel computing on multiple CPU cores. The software includes modules for camera control and image acquisition, preprocessing and data analysis. The total processing time of 3000 12-bit monochrome images with 1200 × 1200 resolution was about 12 min using Intel’s 8th generation 4-core processor. The most time-consuming processes were local matching (18 min) and vessel image calculation (1 min). Further optimization may be related to GPU computing.

Figure 5 illustrates the vessel activity across the whole embryo and the heart rate at the stages from 22 hpf to 27 hpf. Due to the absence of anesthesia, embryos moved and were randomly oriented in chorions. For this reason, the viewing angle and orientation of the embryos differed. Figure 5 shows temporal dynamics of the key features indicating the state of the embryos and suitability for non-invasive measurements at early developmental stages. The experiments demonstrated a continuous increase of the heartbeat rate and gradual activation of vasculature initiated by the heart functioning. Such measurements are important for predicting the further development of the embryo, in particular the time of its hatching.

## 4. Discussion

As a model biological system, the zebrafish possesses numerous advantages: rapid embryonic development, fully sequenced genome, low cost, etc. This organism also has a unique collection of features at pre-larval developmental stages: large size, optical transparency of the embryo’s interior and relatively slow embryo movements inside the shell. This makes zebrafish an attractive model for in vivo study of the formation and functioning of its cardiovascular system using optical imaging techniques.

In this study, we have demonstrated that time-lapse bright-field microscopy and digital signal processing allow blood vessel imaging of a living embryo as well as heartbeat and blood flow velocity measurements without any anesthesia. In contrast to straightforward approaches based on image subtraction, the described pre-processing procedure enables the compensation of shifts and rotations between images and their pixel-to-pixel matching necessary for accurate quantitative characterization of cardiac activity.

The proposed algorithm is applicable for processing the images obtained by various microscope setups widely used for in vivo studies of zebrafish embryos. It can complement optical coherence tomography, acoustic microscopy and other imaging techniques by adding cardiac activity mapping capability.

## 5. Conclusions

The presented algorithm does not require the use of contrast agents and may be used to visualize cardiac morphology and measure dynamics in zebrafish embryos without anesthesia. We believe that it may help to shed light on the mechanisms by which the cardiovascular system’s activity is initiated under natural conditions. Continuous study of the heart function and vessel structure transformation using the proposed algorithm may contribute to recognizing possible warning signs of developmental disorders at the very early stages and understanding the symptoms of various diseases, i.e., help to improve the techniques for their correct diagnosis and timely treatment.

Further development of this technique may include the interactive analysis of vessel structure and function in both normal and pathological condition at embryonic stages, as well as detecting and studying the changes in zebrafish cardiovascular system initiation, development and functioning induced by various environmental stressors.

## 6. Ethics

The authors confirm that all methods were carried out in accordance with relevant guidelines and regulations and were approved by the Lomonosov Moscow State University Bioethics Committee (Protocol #108-0).

## 7. Data Accessibility

The raw image sequence for the 26 hpf unhatched zebrafish embryo is available from Mendeley Data (http://dx.doi.org/10.17632/kj7mkcw8vn.1).

## Figures and Tables

**Figure 1 diagnostics-10-00886-f001:**
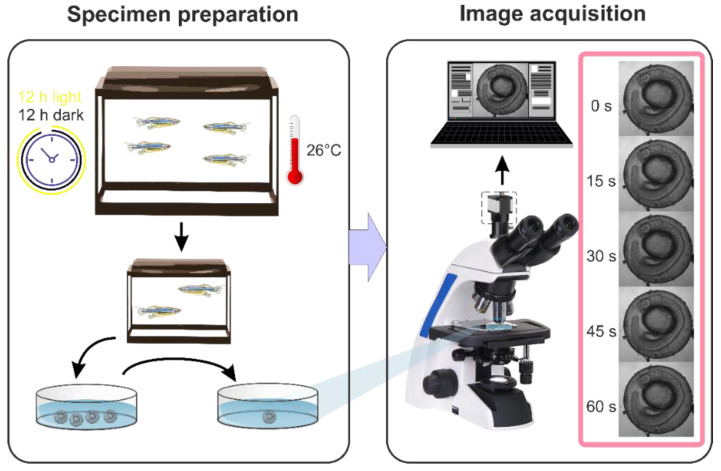
Experimental protocol.

**Figure 2 diagnostics-10-00886-f002:**
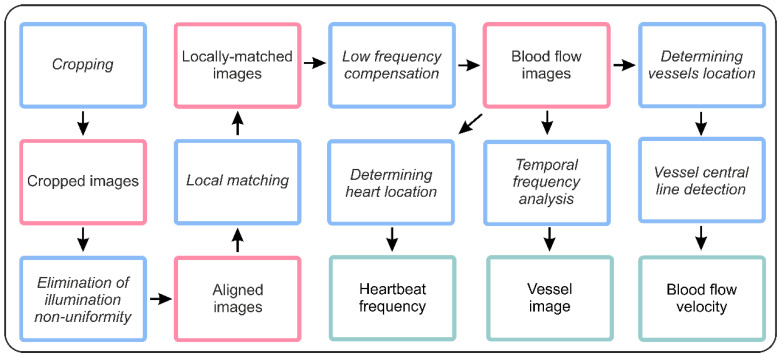
Image processing pipeline.

**Figure 3 diagnostics-10-00886-f003:**
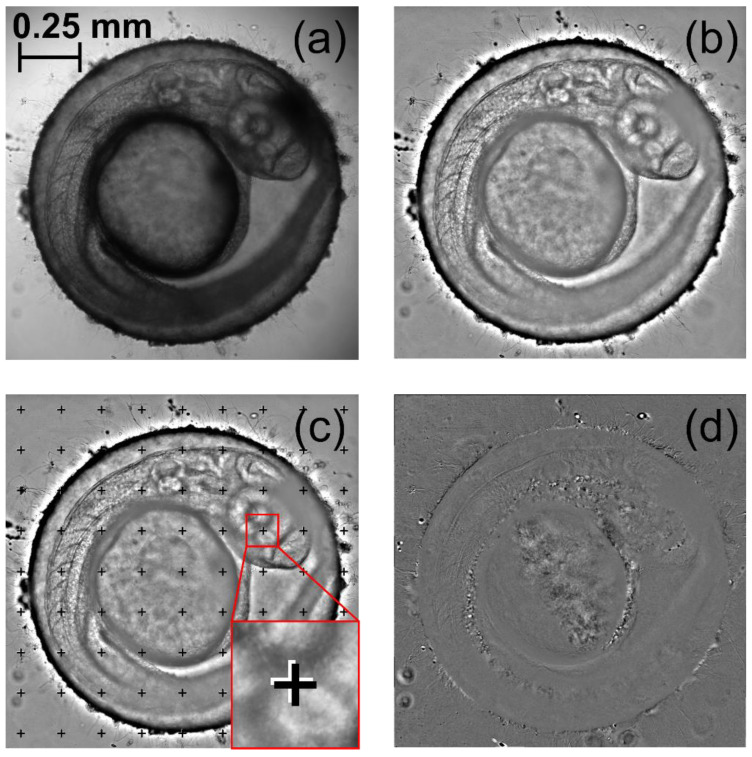
Stages of image pre-processing: (**a**) cropped image, (**b**) aligned image, (**c**) locally matched image and (**d**) blood flow image.

**Figure 4 diagnostics-10-00886-f004:**
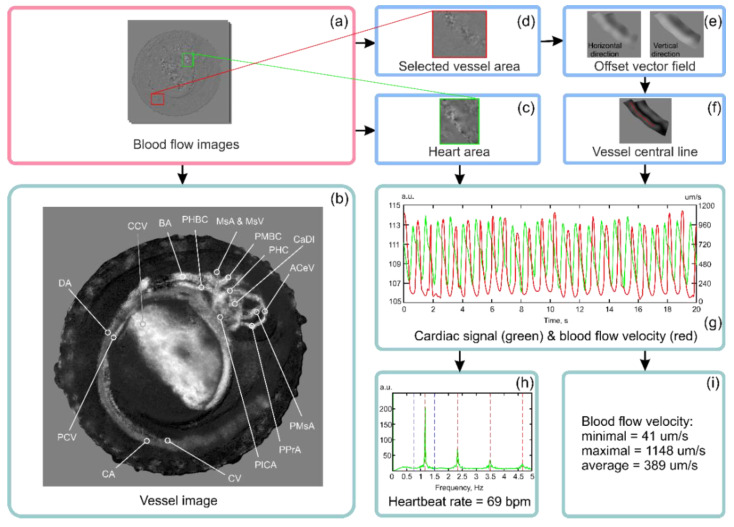
Example of cardiovascular data extraction using the proposed algorithm: (**a**) blood flow images, (**b**) vessel image, (**c**) the selected heart area, (**d**) the selected vessel area, (**e**) offset vector field examples, (**f**) the detected vessel central line, (**g**) cardiac signal and blood flow velocity examples, (**h**) cardiac signal spectrum, (**i**) blood flow velocity range and average value.

**Figure 5 diagnostics-10-00886-f005:**
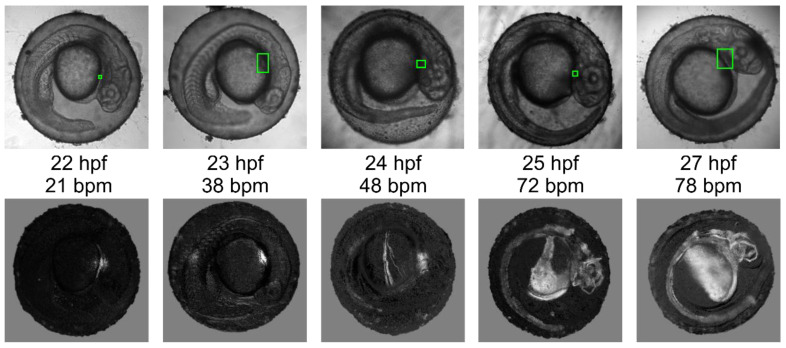
Calculated vessel images (green rectangles indicate the detected heart area).

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
