# Peer review of "Blood Vessel Imaging at Pre-Larval Stages of Zebrafish Embryonic Development"

_diagnostics, 2020, doi:10.3390/diagnostics10110886_

Round 1

Reviewer 1 Report

The authors should explain better the aim and the novelty of the paper. Moreover, the results should be improved adding at least more pictures.

Author Response

Thank you for your valuable comments. To address your remarks, we have revised the manuscript. Changes are marked with green color.

The authors should explain better the aim and the novelty of the paper. Moreover, the results should be improved adding at least more pictures.

The aim and novelty of the paper are stated in Introduction: In this research, we show that combination of time-lapse bright-field microscopy and advanced image processing technique represents an effective approach to in vivo stain-free cardiac activity mapping across the whole specimen and quantitative analysis of cardiovascular performance even at pre-larval developmental stages in the absence of any anesthesia.

To emphasize this, we have expanded Abstract. We have also added Fig. 3 illustrating the proposed image processing algorithm in detail and Fig. 5 demonstrating more experimental results.

Reviewer 2 Report

In the manuscript by Machikhin et al. an image analysis pipeline is described for the analysis of zebrafish embryos at early stages of development with the goal of imaging the development of blood vessels and the blood flow within them.

The main limitation of the paper as it stands is that the analysis method (which is meant to be the main point of the paper) is only qualitatively described, with no demonstration of each step. A methodological paper of this sort would require, in my opinion, a full description of the software and routines/macros used (which should also be made available) with demonstration of the various steps with display of the results instead of a pipeline such as the one shown in Fig.2.

Also, I would recommend reporting some biologically relevant (even very simple) measurement to provide a demonstration of the usefulness of the method in its applications to zebrafish rather than just a proof of concept as it stands.

minor points:

1) correct "shed the light" into "shed light"
2) reference 19 and 26 point at the same paper

Author Response

Thank you for your valuable comments. To address your remarks, we have revised the manuscript. Changes are marked with green color.

(1) The main limitation of the paper as it stands is that the analysis method (which is meant to be the main point of the paper) is only qualitatively described, with no demonstration of each step. A methodological paper of this sort would require, in my opinion, a full description of the software and routines/macros used (which should also be made available) with demonstration of the various steps with display of the results instead of a pipeline such as the one shown in Fig.2.

We have added a description of the software and inserted Fig. 3 illustrating the main steps of the proposed image processing algorithm. We plan to upload the core libraries for public use as soon as we finish optimization of the time-consuming procedures.

(2) I would recommend reporting some biologically relevant (even very simple) measurement to provide a demonstration of the usefulness of the method in its applications to zebrafish rather than just a proof of concept as it stands.

We have added Fig. 5 illustrating the vessels activity across the whole embryo and the heart rate at a few different developmental stages. This is a biologically relevant measurement. It shows temporal dynamics of the key features indicating the state of the embryo and available for non-invasive measurements at such early stages. Further biological studies using the proposed algorithm may be related, for example, to the analysis of the embryonic resistance to various stressors.

(3) correct "shed the light" into "shed light"

Corrected.

(4) reference 19 and 26 point at the same paper

Corrected.

Round 2

Reviewer 1 Report

The paper has been improved following the review suggestions.

Author Response

Thank you for your valuable comments. We were glad to satisfy your review suggestions.